# Flexible genes establish widespread bacteriophage pan-genomes in cryoconite hole ecosystems

Christopher M. Bellas [1]✉, Declan C. Schroeder [2,3], Arwyn Edwards[4], Gary Barker[5] & Alexandre M. Anesio [6]

Bacteriophage genomes rapidly evolve via mutation and horizontal gene transfer to counter evolving bacterial host defenses; such arms race dynamics should lead to divergence between phages from similar, geographically isolated ecosystems. However, near-identical phage genomes can reoccur over large geographical distances and several years apart, conversely suggesting many are stably maintained. Here, we show that phages with near-identical core genomes in distant, discrete aquatic ecosystems maintain diversity by possession of numerous flexible gene modules, where homologous genes present in the pan-genome interchange to create new phage variants. By repeatedly reconstructing the core and flexible regions of phage genomes from different metagenomes, we show a pool of homologous gene variants co-exist for each module in each location, however, the dominant variant shuffles independently in each module. These results suggest that in a natural community, recombination is the largest contributor to phage diversity, allowing a variety of host recognition receptors and genes to counter bacterial defenses to co-exist for each phage.

[1] Department of Ecology, Lake and Glacier Ecology, University of Innsbruck, Technikerstrasse 25, 6020 Innsbruck, Austria. [2] Department of Veterinary Population Medicine, University of Minnesota, 1333 Gortner Avenue, St. Paul, MN 55108, USA. [3] School of Biological Sciences, University of Reading, Reading, UK. [4] Institute of Biological, Environmental and Rural Sciences, Aberystwyth University, Aberystwyth SY23 3EE, UK. [5] School of Biological Sciences, University of Bristol, 24 Tyndall Avenue, Bristol BS8 1TQ, UK. [6] Department of Environmental Science, Aarhus University, Frederiksborgvej 399, 4000 Roskilde, Denmark. ✉email: christopher.bellas@uibk.ac.at

The interactions of viruses and their hosts have large-scale consequences for global carbon and nutrient cycling[1,2], microbial population dynamics[3] and the use of bacteriophages as antimicrobials[4]. Bacteriophage–host interactions in nutrient-rich culture often follow arms race dynamics, where successive generations of bacteria become increasingly resistant to past phages, and in turn, phages become more infective to previous bacterial generations, leading to a relative balance between contemporary phages and hosts[5]. At the molecular level, one of the genes involved is the phage tail fibre gene, where single nucleotide variations between each generation significantly change a phage's host range[6]. If arms race dynamics continue unimpeded, simultaneously in geographically isolated ecosystems, this will produce divergence between many phage genes responsible for host recognition and countering bacterial defences and create an endless variety of mutations. However, in environmental studies, arms race dynamics appears to be more limited, giving way to fluctuating section on longer time-scales which can potentially continue indefinitely[7].

When bacteriophage and prokaryotic reference genomes are used to recruit metagenomic sequencing reads from their natural environment, there are often gaps in the alignment where regions of the genome significantly under-recruit, or fail to recruit any reads, while the rest of the genome is uniformly covered by reads matching at close to 99% nucleotide identity[8–12]. These regions were described as Metagenomic Islands (MGIs) in bacteria[12] and later Metaviromic Islands in viruses[10], however we use the original term MGI to denote both occurrences in this study, as these regions occur in viruses present in both metagenomes and metaviromes (<0.2 μm size fraction). MGIs are suggestive of highly variable genes present in co-existing variants of the same species, the pan-genome. The pan-genome describes the total compliment of genes present in a species, consisting of a core genome shared amongst all representatives and a flexible genome within that species[13]. This is perhaps best exemplified in the abundant marine cyanobacterium Prochlorococcus, where the core genome makes up approximately half of all genes, with the other half (the flexible genome) being shared by only a subset of isolates[14]. Many of these variable genes cluster together in genomic islands, often including genes involved in lipopolysaccharide or cell wall biosynthesis, which are known to alter the susceptibility of the cell to bacteriophages[15,16]. Hence, co-existing lineages of prokaryotes, encoding varied cell surface receptors, may reduce the predation pressure they receive from a particular phage. To counteract this, it is proposed that phages co-exist as a population of near identical variants, each encoding a variety of host recognition receptors to infect each bacterial variant. This diversity generating scenario has been termed the constant diversity model (CD model)[12]. Both CD and divergence via arms race dynamics could produce the observations of MGIs in phages[10]. These are not mutually exclusive processes, however arms race dynamics should lead to unique variation in both phage and host genes in each isolated ecosystem, with presumably increased genomic variation over time and large spatial scales. CD should produce multiple variants in each location, with a change in individual variant frequencies causing the observed MGIs. The evolutionary implications are that a CD of homologous phage gene variants would promote a more diverse population of bacterial variants in each location[17].

There is much evidence that the same marine phage genomes reoccur seasonally[18,19], across large distances and even between biomes[20,21]. Marine phage genome stability has been confirmed using large genome fragments monitored for over a year[22] and over decades in complete marine myoviruses genomes[23]. Similarly, large-scale metagenomic comparisons have also revealed persistent virus types across global scales, with genomes widely dispersed across the oceans[24,25]. Such occurrences are paradoxical when considering the known, rapidly evolving nature of phage genomes and extensive horizontal gene transfer[26,27]. Understanding how phages maintain this apparent stability is a major question in phage genomic diversity.

To investigate stable virus genomes in widespread environmental samples and their connection to MGIs, we analysed phage genomes across cryoconite hole ecosystems. Cryoconite holes are unique hot spots of microbial diversity and activity on glaciers[28–31]. They are small water filled depressions, typically a few tens of centimetres wide and deep, with a thin (2–20 mm) dark layer of mineral and biological material at the bottom which is bound together by abundant filamentous cyanobacteria[32]. This dark sediment warms in the sun and melts into the ice, forming the pool of water above. Cryoconite holes possess abundant and active virus communities, with circa $10^8$ virus-like particles (VLP) $g^{-1}$ of sediment and new virus production rates in the order of $10^7$ VLP $g^{-1}$ $h^{-1}$ [33]. A broad range of novel viruses have been described in these habitats[34,35]. Across Earth's glaciated regions, cryoconite holes typically experience near-freezing temperatures, nutrient limitation and possess a highly truncated food web[36]. They can be hydrologically connected, where melt water moves through the top layer of ice, however the sedimentary material itself, where the vast majority of microbial activity takes place, remains on the ice for years to decades[37]. Cryoconite holes may therefore be considered as small, naturally occurring microcosm-like ecosystems. In this study, we took advantage of their unique and consistent nature to recruit metagenomic reads to virus reference genomes and genome fragments from Svalbard, Greenland and the Alps. The relatively low diversity of these ecosystems allows for repeated detection of the same virus genomes in multiple metagenomes, with sufficient read depth to identify MGIs. We identify the most abundant categories of virus genes present in MGIs, and in two bacteriophages with sufficient coverage, we reconstruct the variable genes present in multiple locations to determine their relationships and the processes causing observed MGIs in viruses.

## Results

**Near-identical bacteriophage genomes**. To identify widely distributed phage genomes we used a dataset of 671 virus genome and genome fragments assembled from Svalbard, Greenland and Alpine cryoconite (Table 1). By mapping metagenomic reads to the reference genomes (>90% identity) (see "Methods" section) we identified 257 viruses (38%) that were present in two or more regions (Fig. 1), with 50 detected in all three regions. As almost all of our reference database was composed of phages, these viruses were the biggest contributors to the MGI dataset, hence results will be discussed in terms of phages from this point forwards, however three nucleocytoplasmic large DNA virus (NCLDV) genome fragments also occurred in more than one location and exhibited MGIs (Supplementary Data 1).

**MGIs are present in most phage genomes**. In total 2612 MGIs were detected in phage genomes based on drops in read-mapping coverage (see "Methods" section), with a mean MGI size of 448 bp (range 100 bp to 7 kb) (Fig. 1). MGIs were found in 82% of phage genomes when recruiting reads from another location, with ~10% of each reference genome composed of MGI regions. Where the same phage was detected in three or more locations, MGI regions usually occurred in the same gene (Figs. 2 and 3). In total 3083 genes were predicted to overlap with ≥10% of an MGI. Of these, 482 (16%) could be assigned a function based on homology searches against GenBank, Pfam and UniProtKB databases (Table 2; Supplementary Data 1). The most frequently

**Table 1 Metagenomes and virus genome/fragments used in this study.**

| Sample | Size (GB) | Location (Glacier) | Year | Type | Accession number | Virus contigs >15 kb | Ref. |
|---|---|---|---|---|---|---|---|
| Green | 8.3 | Greenland (Russell Glacier) | 2010 | V | SRR8842250 | 268 | 34 |
| ML09 | 9.6 | Svalbard (Midtre Lovénbreen) | 2009 | V | SRR8842249 | 151 | 34 |
| AB09 | 6.9 | Svalbard (Austre Brøggerbreen) | 2009 | V | SRR8842248 | 128 | 34 |
| ALP | 27.4 | Alps (Rotmoosferner) | 2009 | MG | mgm4491734.3 (MG-RAST) | 124 | 28 |
| ML10 | 5.3 | Svalbard (Midtre Lovénbreen) | 2010 | MG | SRR12327455 | – | This study |
| GreenM | 5.1 | Greenland (Russell Glacier) | 2010 | MG | SRR12327363 | – | This study |
| ML13 | 8.7 | Svalbard (Midtre Lovénbreen) | 2013 | MG | SRR12350504 | – | 70 |

V—virus enriched metagenome (<0.2 μm) from ~500 g sediment of pooled cryoconite; MG—metagenome from 0.25 g pooled sediment, ML10, GreenM and ML13 used for MGI reconstructions only in Fig. 2 and 3.

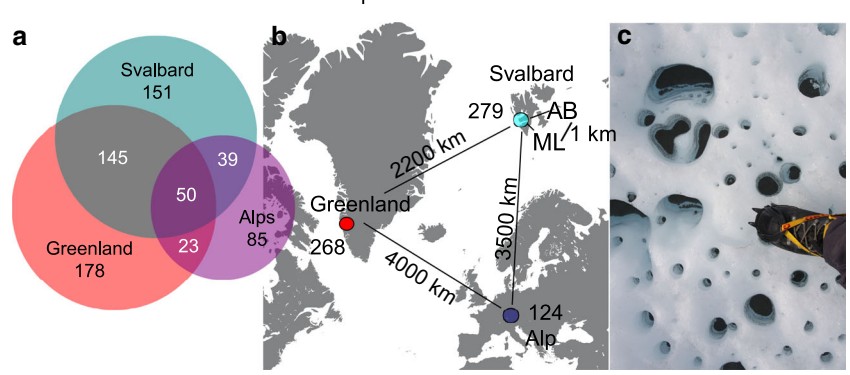

Virus scaffolds over 15 kb in multiple locations

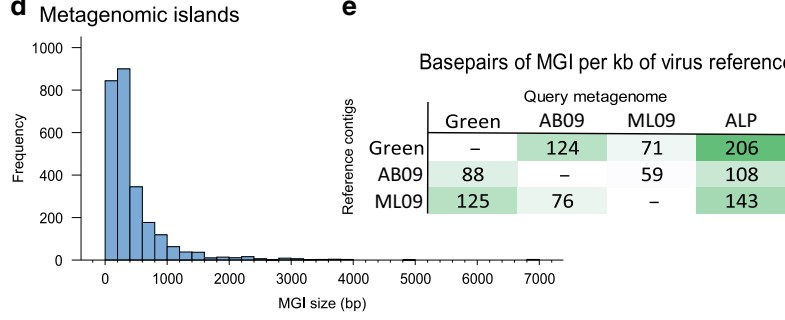

**d** Metagenomic islands

**e** Basepairs of MGI per kb of virus reference

**Fig. 1 Cryoconite virus recruitment and metagenomic islands.** Panel **a** shows the number of virus contigs that were detected in a particular region via metagenomic recruitment and the number detected in two or more locations which were used for MGI detection; Panel **b** shows the number of >15 kb virus contigs assembled from each location; Panel **c** shows typical cryoconite holes from Greenland with the authors boot for scale (Photo credit: C. Bellas); Panel **d** is the frequency distribution of all detected MGI lengths; Panel **e** shows a heat map of the mean base pairs of MGI per kb of virus reference genome for each read mapping combination. AB Austre Brøggerbreen (Svalbard), ML Midtre Lovénbreen (Svalbard), Green Russell Glacier (Greenland Ice Sheet), ALP Rotmoosferner (Austria). Source data are provided as a Source Data file.

occurring genes found in MGIs were orphan DNA Methyltransferases (MTases), accounting for 11% of the known MGI genes ($n = 51$). MTases protect phage genomes from specific host encoded restriction modification (RM) systems[38,39]. Their high frequency in MGIs suggests they are variable or mobile genes which protect phage variants against different host encoded RM systems. Mobile elements, such as homing endonucleases, transposases and inteins were the next most frequent group of genes (10% of all MGI genes). Homing endonucleases recognise and cleave a specific DNA sequence, inserting their own gene into a recognition site which lacks the gene. They replicate at little cost to their hosts[40] and may be considered selfish genetic elements. However, some phage encoded HNH homing endonucleases can competitively exclude co-infecting phages by preferentially cleaving their DNA[41]. Hence, the highly variable nature of

homing endonucleases between phage variants here may be simply the movement of selfish genes, or a mechanism to out compete co-infecting phages in a host.

A large proportion of annotated MGI genes (19%) appeared to be involved in host recognition and cell entry (Supplementary Data 1). This includes putative receptor-binding proteins such as phage tail-related gene hits (tail fibre/tail protein/collar/sheath, $n = 36$) and also C1q-like domains ($n = 3$) which are pattern recognition receptors frequently found in marine phage MGIs[10]. Furthermore, putative polysaccharide depolymerases and virion-associated lysins ($n = 50$) were detected. These are highly specific recognition proteins which degrade bacterial surface components, such as an exopolysaccharide (EPS) and allow phages to puncture their hosts' carbohydrate barriers[42,43]. Putative polysaccharide depolymerases and lysins found in MGI

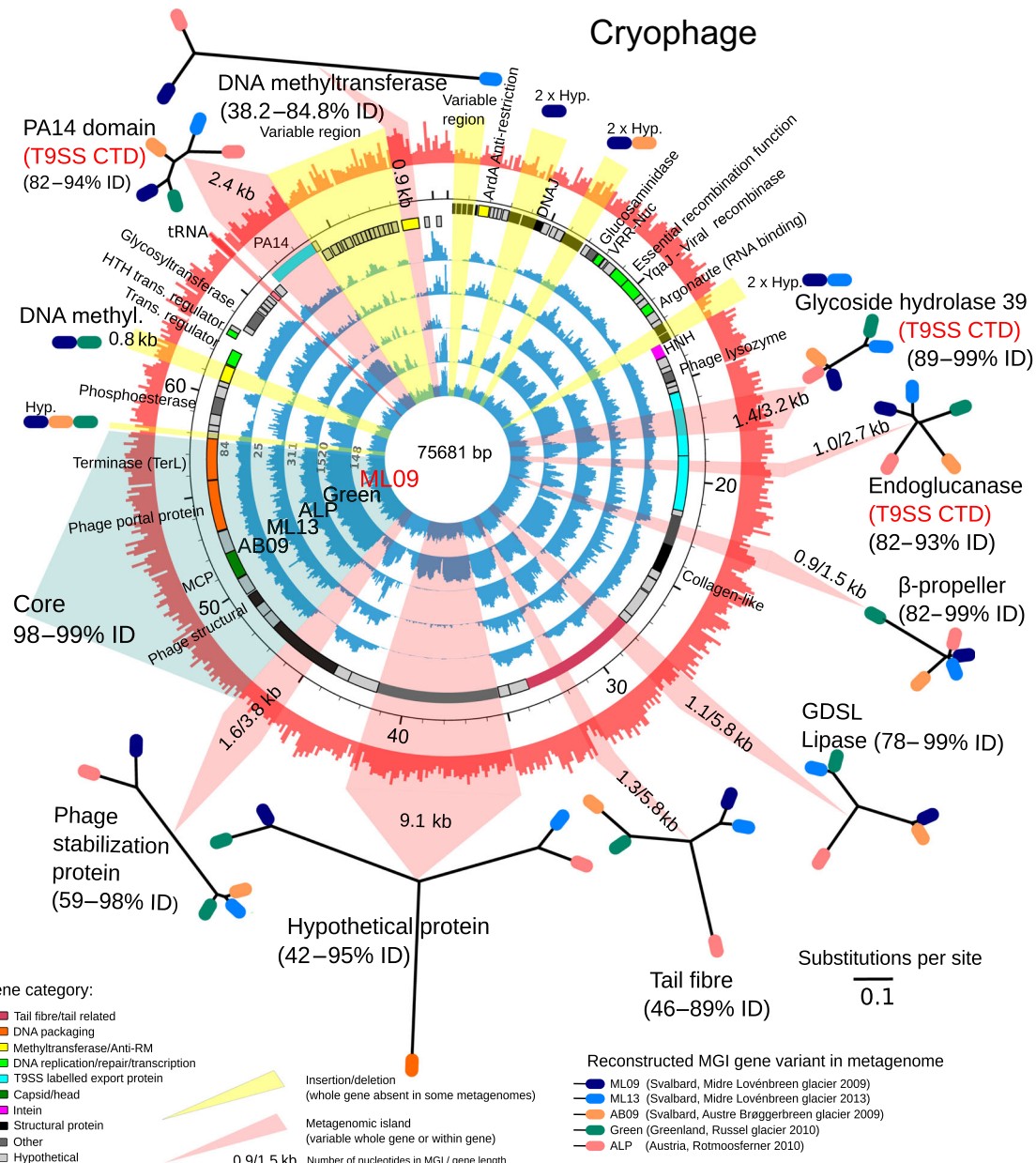

**Fig. 2 Cryophage recruitment and genome reconstructions.** Inner five circles represent read recruitment of each metagenome (coverage scale indicated in grey of the left). Sixth circle from centre represents gene predictions and annotation, outer circle represents the GC plot. The read recruitment plot was generated with Circleator[73]. The phage was originally assembled from the ML09 metagenome, MGIs which arise from gene insertion/deletion events are highlighted in yellow overlay, the coloured ovals denote the metagenomes where the insertion is present. MGIs derived from variable genes are highlighted in pink. Reconstructions of the variable regions for each metagenome are visualised in maximum likelihood trees generated from nucleotide alignments, the coloured ovals represent which metagenome the gene variant was reconstructed from. The minimum and maximum percentage identities between all MGI variants are shown under the label. GDSL—GDSL Lipase, Hyp—hypothetical protein, MCP—major capsid protein, T9SS CTD—Type-IX secretion system labelled cargo proteins. Source data are provided as a Source Data file.

regions in this study include pectate lyases, peptidases, endopeptidases, glycoside hydrolases, and GDSL-lipases (Supplementary Data 1). The frequent occurrence of genes involved in host recognition in phage MGIs suggests phages are adapting to infect a range of host strains with diverse EPS, cell surface receptors and carbohydrate barriers.

Many of the other genes in MGIs (19%) were involved in DNA replication and repair, particularly DNA polymerase and ribonucleotide reductase genes (RNR) (Table 2; Supplementary Data 1). Approximately half of these genes were detected as "additive" MGIs where the whole gene, or an insertion sequence

within, is deleted relative to the reference genome, hence the gene or sequence could be present/absent but not variable. DNA polymerases and RNR genes are known to contain a disproportionally high frequency of homing endonucleases[44], hence insertions and deletions of mobile elements may cause the MGIs detected in association with these genes. Indeed, three of 11 RNR genes in MGIs possessed homing endonucleases, with two more showing the RNR gene was split by an unknown insertion sequence. However, complete DNA polymerase and RNR genes were also observed as whole gene insertions/deletions in MGIs. RNRs are one of the most common auxiliary metabolic genes

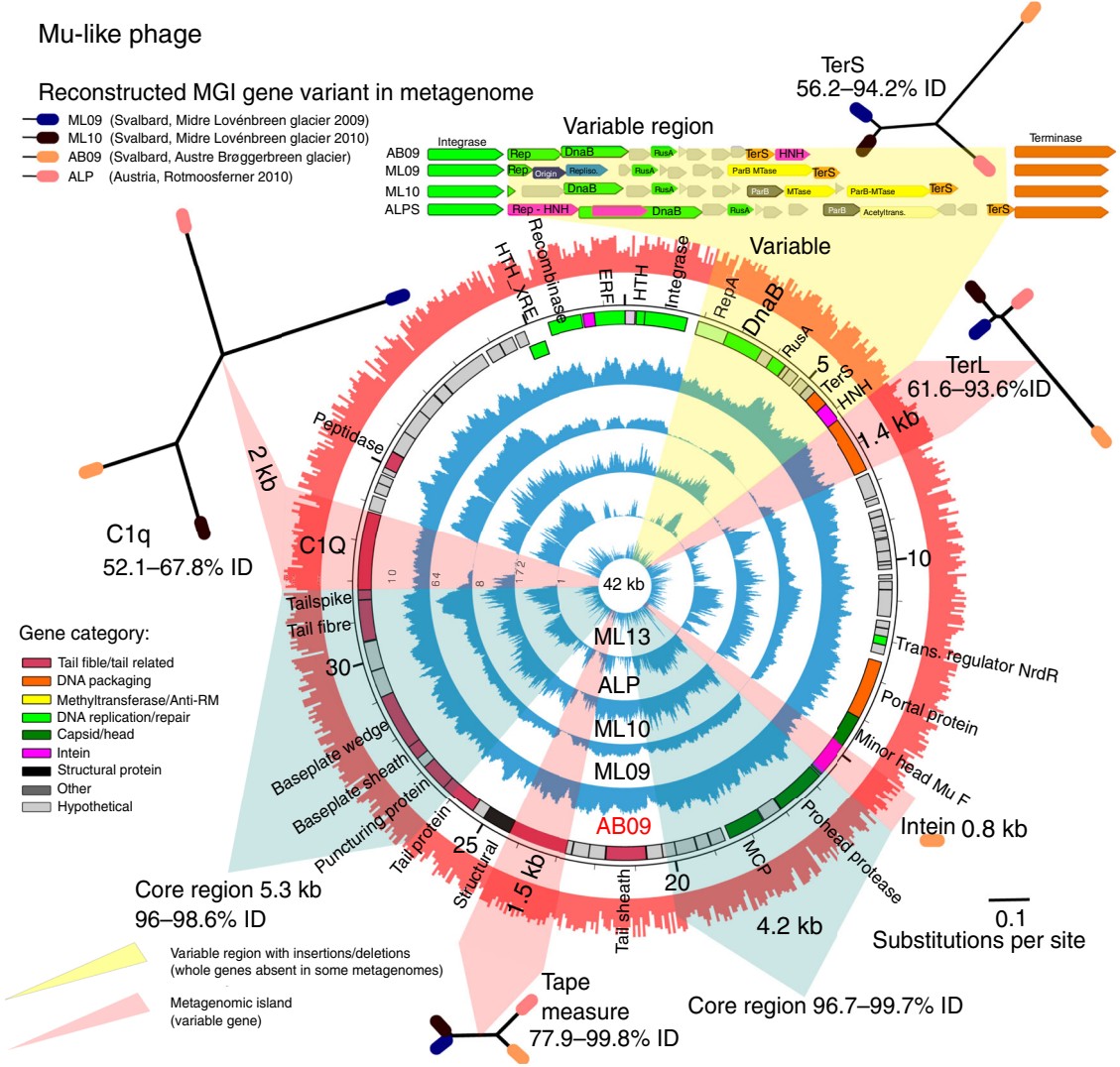

**Fig. 3 Mu-like phage recruitment and genome reconstructions.** The phage was originally assembled from the AB09 metagenome, the layout is as per Fig. 2 with variable metagenomic islands reconstructed in each metagenome and visualised in maximum likelihood trees based on nucleotide alignments of whole genes. Two core genome regions, highlighted in blue, show 96–99% nucleotide identity. A variable region is present between the phage integrase gene and large terminase subunit. RusA—RusA family crossover junction endodeoxyribonuclease, Rep—replication initiator, ParB—ParB domain protein nuclease, TerL—large terminase subunit, TerS—small terminase subunit, C1q—C1q-like domain and putative receptor-binding protein. Source data are provided as a Source Data file.

encoded by phages, reducing ribonucleotides to deoxyribonucleotides and useful for phage taxonomy[45]. Terminase genes involved in DNA packaging were also among the most frequently observed genes in MGIs (6% of MGI genes), however, the role of their variability remains unknown.

Our analysis of MGI regions in cryoconite phage genomes is largely in agreement with a study on marine phages from a single location, which detected MGIs in genes involved in host recognition, including carbohydrate binding, tail fibre and C1q domains, but also small and large Terminase subunit genes[10]. Our results differed however, in that DNA methyltransferases and homing endonucleases were the two most frequently variable/mobile phage genes in the cryoconite dataset, in the latter case this was most likely owing to the inclusion of smaller MGIs in our dataset.

**MGIs represent flexible, homologous genes**. To determine the new DNA sequence present in gapped MGI regions, we developed an in-silico approach (see "Methods" section) to close the gaps in phage genomes using metagenomics reads. We chose two,

widespread phages with numerous MGI regions that were present in five or more metagenomes, these had sufficient coverage depth (>10× coverage) to allow reconstruction of the gapped MGI regions. As MGIs occurred in all phage types in the dataset, the results of this analysis are not specific to these two phages.

The first genome was a novel, circular, 76 kb Bacteroidetes phage genome (which we refer to as Cryophage owing to its omnipresence in Cryoconite) which was assembled/reconstructed in full in five distant metagenomes, the second was a 42 kb Mu-like phage genome which was assembled/reconstructed in full in four metagenomes, both exhibited multiple MGI gaps in the read mapping. Cryophage recruited up to 1% of all metagenomic reads (ca. 1 million reads) in the Greenland virus metagenome. We infer the host as 22 of 34 BLASTP (e-value < 10−5) gene hits, including the major capsid protein, are to integrated elements in the Phylum Bacteroidetes (Supplementary Data 2). A 10 kb consecutive nucleotide alignment in a core genome region of all Cryophage variants demonstrated that the genomes shared 99% nucleotide identity between locations. Reconstructions of the missing DNA sequences in the MGIs confirmed they contained

**Table 2 Most frequently occurring known gene types overlapping with an MGI.**

| Annotation | Hits | Del | Intein | Known bacteriophage function | Ref. |
|---|---|---|---|---|---|
| DNA methyltransferase | 51 | 9 | 0 | Anti-restriction modification | [71] |
| Homing endonucleases (all) | 38 | 13 | 17 | Mobile element | |
| Tail fibre/collar/protein | 36 | 4 | 0 | Host recognition | [42] |
| Terminase (TerL/TerS) | 16 | 4 | 1 | DNA packaging | |
| Ribonucleoside reductases | 13 | 5 | 3 | DNA replication/repair | |
| DNA polymerases (all) | 13 | 5 | 0 | DNA replication/repair | |
| Structural proteins (all) | 11 | 0 | 0 | Structural | |
| Recombinase | 9 | 2 | 0 | DNA replication/repair | |
| GDSL Lipase | 8 | 0 | 0 | Cell entry/exit | |
| Peptidase (all types) | 8 | 0 | 0 | Increase phage adsorption/Endolysin | [42] |
| Nucleases (endo/exo) | 8 | 2 | 0 | DNA replication/repair | |
| Transcriptional regulator | 8 | 0 | 0 | Transcriptional regulator | |
| Glycosyltransferases (all) | 6 | 1 | 0 | Superimmunity/Anti-RM | [72] |
| Repressor/antirepressor | 6 | 0 | 0 | Transcriptional regulator | |
| Integron gene cassette | 5 | 0 | 0 | Mobile element | |
| Transposase | 5 | 2 | 0 | Mobile element | |
| Pectate lyase | 5 | 0 | 0 | Host recognition of EPS | [71] |

Del—number of MGIs bridged by >10% discordantly mapping reads (a deletion event between locations); Intein—number of MGIs with a database hit to a known intein domain (CDD search).

highly diverged sequences of homologues genes. Each reconstructed MGI nucleotide sequence exhibited a different relationship between metagenomes (Fig. 2), with identical variants often found thousands of kilometres away, yet replaced by alternative variants 1 km away. For example, in the large open-reading frame containing the tail fibre and GDSL lipase domains, both regions exhibited MGIs (Fig. 2). Reconstructions of the lipase domain in each location show three distinct variants are present across the five independent metagenomes: ML13 and Green metagenomes share a single variant (99% nucleotide identity) between locations, whilst AB09 and ML09 metagenomes share another (99% identity) whilst the ALP variant was distant to all (~46% identity). However, in the tail fibre domain five distinct variants were found with <90% nucleotide identity in the five metagenomes. A large 7 kb open-reading frame (designated hypothetical protein) shows a different relationship again and is distributed into three highly dissimilar groups sharing as little as 42% nucleotide identity (ALP vs. ML09). A 0.8 kb DNA MTase is present in ML09 at position 61 kb, but is completely absent in ALP, ML13 and AB09, yet present in Green at ~50% of the read recruitment coverage. Another 0.9 kb DNA MTase is present in three distinct variants sharing as little as 38% identity at position 74 kb in ALP, ML09 and ML13. Similarly, the other reconstructed MGI variants exhibited a different relationship between the geographic locations (Fig. 2). Three variable genes in Cryophage also encoded Type IX secretion system (T9SS) type A sorting domains: an endoglucanase, a glycoside hydrolase type 39 and a PA14 domain containing gene, often found in bacterial toxins and signally molecules[46]. The T9SS is a recently discovered secretion system which translocates labelled proteins (often virulence factors) with a conserved C-terminal domain (T9SS CTD) across cell outer membranes[47]. It is exclusive to the Fibrobacteres-Chlorobi-Bacteroidetes superphylum[48]. Cryophage therefore represents a unique occurrence of phage-encoded T9SS cargo proteins, which presumably hijack the host encoded system to secrete virulence factors or provide additional enzymatic capabilities, but most interestingly here, the cargo proteins themselves are highly variable between Cryophage variants. This suggests that Cryophage may cause its host to secrete a variable complement of membrane-associated proteins or virulence factors depending on the exact variant infecting the cell, which may be involved in inter-strain competition or superinfection exclusion.

In the Mu-like phage genome, a similar pattern emerged in the gene reconstructions. The phage genome possessed a near-identical gene arrangement between the four reconstructed variants, aside from a 2 kb variable region in the DNA packaging module where DNA MTases were inserted and deleted, and a 2 kb region where two homing endonucleases were inserted into Rep and DnaB proteins (Fig. 3; Supplementary Data 2). The core genomes shared 96–99.7% nucleotide identity with the original variant from AB09. MGIs were present across a 1.7 kb gene downstream of the tail fibre encoding a putative receptor-binding protein (C1q domain). Analysis of the four gene variants containing C1q domains revealed a distinct variant in each location, sharing 54–69% nucleotide identity (Fig. 3). In the tail tape measure gene, three variants were present, two phage variants from the same glacier (but consecutive years) shared near-identical genes (ML09 and ML10—99.8% nucleotide identity), whilst the AB and ALP variants shared 78% and 81% identity to the original variant. In the DNA packaging module, the large terminase subunit (TerL) consisted of two similar variants; ML09, ML10 and ALP shared a similar TerL gene (90–94% identity), which in turn each shared only ~60% identity with the original AB09 terminase. DNA MTases were highly variable, ML10 encoded two DNA MTases, ML09 encoded one, whilst the ALP variant encoded an acetyltransferase gene in the same region. Finally, an intein created MGIs in the AB09 minor head protein (0.8 kb homing endonuclease).

**Multiple gene variants are co-present in each metagenome.** By recruiting each metagenome back to all MGI gene variants in Cryophage and the Mu-like phage (see "Methods" section), we determined that for an individual MGI gene, up to five variants were co-present in each metagenome (Fig. 4). Of note, is the C1q-like putative receptor-binding protein in the Mu-like phage where three of four distinct variants were co-present in the ML10 metagenome. Similarly, in Cryophage, all five tail fibre domain variants were co-present in the GreenM metagenome (assuming >90% nucleotide identity to each reference variant). This implies that the MGI gene variant reconstructed for each metagenome (Figs. 2 and 3) represents the most abundant variant at the time of sampling, however, the relationships between widely dispersed phages are shuffled for each MGI, and the number of variants co-present changes between each MGI and metagenome (Fig. 4). The widely variable MGIs nucleotide sequences, often as low as ~40%

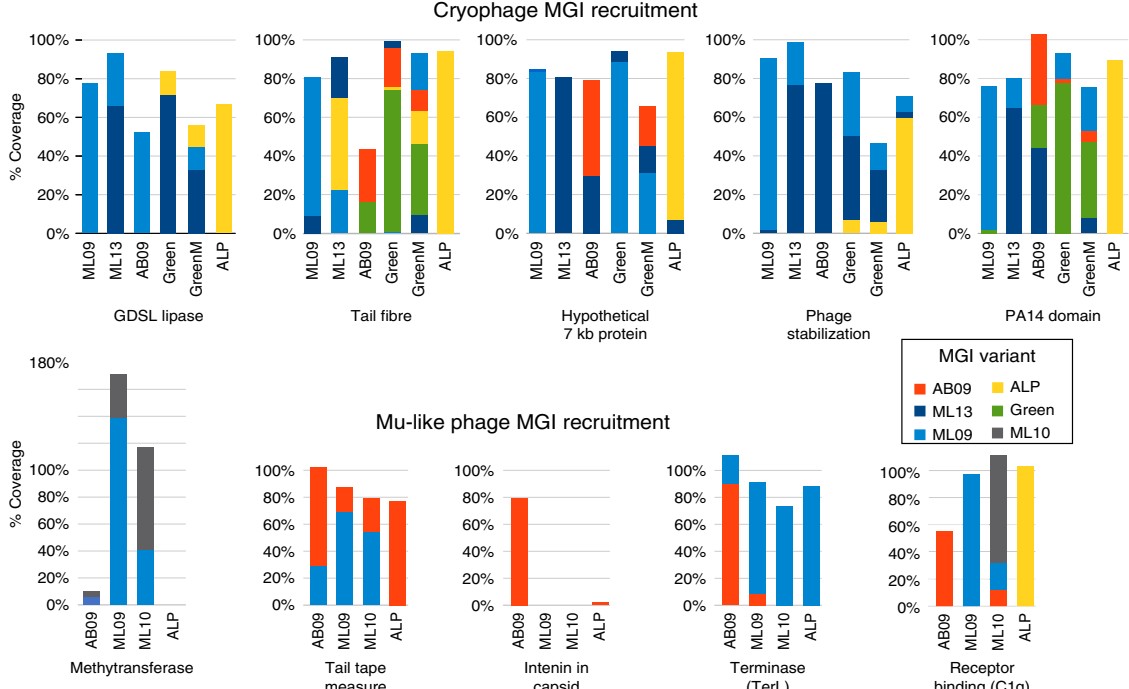

**Fig. 4 Metagenomic recruitment to ten MGI regions.** Each metagenome (x-axis) is mapped to all MGI gene variants and the mean recruitment coverage of each variant is expressed as a percentage of the core genome coverage (y-axis). Different variants share <90% nucleotide identity. The colour and key denote the metagenome where the MGI variant was first assembled or reconstructed if the reconstructions proceeded from left to right, e.g. in the Cryophage GDSL lipase domain the ML09 original variant (light blue) is the only one detected in the ML09 metagenome, whilst in ML13 a new GDSL variant was reconstructed (dark blue) which recruits reads to ~60% of the core genome coverage which are co-present with the original ML09 variant (~20% of core genome coverage). In metagenome AB09 the original ML09 GDSL lipase was the only variant present at 50% core coverage. In the Green metagenome the ML13 variant was independently reconstructed a second time (98.2% identity) accounting for ~70% of core genome coverage, with an additional ~10% recruiting to the ALP reconstructed variant (yellow). Source data are provided as a Source Data file.

nucleotide identity in a core genome which shares up to 99% identity, and the recurrence of near-identical genes, but in different combinations, in isolated phage variants implies recombination events are responsible for the observed variability. Such events are well documented in cultured phages, including between tail fibre genes which alters the phage host range[49,50], in DNA MTases[51], in phage terminase genes (TerS, TerL), tailspike and receptor-binding proteins[52]. In a study of *Lactococcus lactis* 936 group phages from a single dairy over 30 years, recombination was shown to out-compete mutation by a factor of 24 in generating nucleotide variations[53]. A stable core phage genome was maintained over 30 years, with a variable compliment of genes—including phage structural proteins, HNH homing endonucleases, phage lysins and DNA polymerase genes—acquired and switched by recurrent recombination from the phage pan-genome.

## Discussion

Cryoconite holes are ideal locations to test hypotheses in microbial evolution. They are semi-isolated, identical habitats, containing near-identical virus communities. As they exist across large areas, such as the melt zone of the Greenland Ice Sheet, diversity can be tested in these microcosms across spatial scales from centimetres to thousands of kilometres. Here we show that by using a single phage reference genome to recruit metagenomic reads from distant cryoconite holes and backfilling the gaps in silico, we can infer the variable genes present in a population of phages (the pan-genome) and determine the underlying mechanisms generating diversity. Such a technique can be applied when metagenomes are generated from the same location as an isolated reference genome, or when using metagenomically

assembled phage genomes. This technique could also be useful in identifying virulence factors present in a phage pan-genome when only a single reference is known, or identifying all the variants of a novel enzyme with biotechnological applications.

The large proportion of variable genes found in phage genomes with potential host recognition and DNA methylation roles suggests that these regions are involved in phage adaptation to diverse bacterial hosts. Given that single point mutations in receptor-binding proteins can alter the host range of a phage[6], the very low identities described here between MGI gene variants (as low as 46% nucleotide identity) suggest the ability to infect host strains with highly varied cell surface receptors. The ability of bacterial strains to produce a variety of external polysaccharides, including the highly variable O-antigen, is well known and often suggested to be a defence against phage predation[16,54–57]. Hence, diverse co-occurring phage genes involved in host recognition are likely targeting these diverse receptors. Our data is therefore consistent with the CD model[12,17], where a population of near-identical phages are proposed to co-exist in an ecosystem to target different host lineages. From our analysis, the individual phage variants are not truly fixed, but create additional diversity through interchanging genes at specific locations via recombination from a widespread gene pool. The CD model and arms race dynamics are not mutually exclusive, both processes may be occurring simultaneously, as arms race dynamics are well documented in laboratory settings. However, on the large spatial and temporal scales analysed in this study, the detection of stable genomes with a similar complement of interchangeable gene variants, thousands of kilometres apart, suggests arms race interactions do not create unlimited diversity in each location. As we repeatedly detect similar (>90% nucleotide identity) phage tail

fibre variants and receptor-binding proteins in the pan-genome in different locations, it may be that selection for mutations via arms race dynamics fluctuates around each relatively stable gene variant.

The phage genomes we detected in our study were highly stable between distant locations, where they possessed nearly identical nucleotide sequences across >90% of their genome length. However, in the short, variable parts of the genome, the gene switching patterns we observe can only come from recombination which generates microdiversity in the phage population. Such microdiversity, in specific genes, may largely explain the para-doxical, long-term maintenance of stable phage populations across the oceans[20,21,25,58]. Stable phage genomes are often detected based on the presence of overlapping sections of a genome or short fragments, which would not consider highly divergent single genes. Only when complete genomes or large fragments are compared, as in our study, does this microdiversity become apparent. In a study comparing phage genomes from a single sample in the Western Mediterranean[17], many nearly identical genomes were found which had significantly diverged only at specific points in the genome, which were often host recognition genes. Similarly, complete, near-identical marine cyanophage genomes have been shown to reoccur over a decade apart in a study of the NE Atlantic, however, the majority of near-identical genomes (<98.5% average nucleotide identity across the entire genome) were found to contain one or more flexible genes which created diversity at specific points between variants, par-ticularly in tail-related genes and those involved in RM systems[23]. Our results expand on this to show that these diversity-generating regions occur in 80% of phage genomes from cryoconite, across a complete community of phages. We show that recombination can be recurrent in each flexible region, as all gene variants can co-exist in each geographical location. The diversity generating benefits of such a stable core and pan-genome are particularly exemplified by the allelic variation present in Cryophage, an abundant, stable, Bacteroidetes phage found in every cryoconite metagenome. Cryophage draws many parallels with crAssphage, another highly stable, abundant, Bacteroidetes phage genome found globally in the majority of human gut microbiomes[59,60]. crAssphage also exhibit multiple MGIs across its genome between microbiomes[59], suggesting the stability in the human gut microbiome is also maintained by recombination in specific genes from a constantly diverse phage population. In the case of Cryophage, the multiple, interchangeable gene variants likely allow for rapid adaption to changing host populations, for if each of the nine flexible Cryophage genes possesses 2–5 interchange-able, allelic variants, then this allows for over 100,000 unique possible Cryophage combinations before even considering insertion/deletion MGI genes or finer scale evolution via single nucleotide variations.

## Methods

**Virus reference genomes and genome fragments**. Virus genomes and large genome fragments from cryoconite sediment (<0.2 μm size fraction) from Svalbard and Greenland were previously sequenced, assembled from metagenomes and filtered[34] (*n* = 546) (Table 1). Briefly, cryoconite sediment was sampled from the surface of Midtre Lovénbreen (ML) and Austre Brøggerbreen (AB) glaciers near Ny-Ålesund, Svalbard (78°55′N 11°55′W) and from the margin of the Greenland Ice Sheet, near Kangerlussuaq (67°9′39.7′′N, 50°0′52.7′′W). Cryoconite was frozen at −20 °C for return to the laboratory. Virus were separated from the cryoconite by mixing 500–700 g of sediment with phosphate buffered saline, before centrifuging out larger particles and removing bacteria by 0.2 μm filtration. Viruses were concentrated by FeCl₃ precipitation[61] before DNA was extracted by using a QIAamp MinElute Virus Spin Kit (QIAGEN). The three samples were sequenced in half a lane on an Illumina GAII, with 100 bp paired-end reads and an insert size of 400 bp at the Bristol Genomics Facility. Metagenomes were assembled using CLC Genomics Workbench 6. Two further cryoconite metagenomes, ML10 and GreenM, were generated by pooling cryoconite from a 10 m radius at each glacial site and extracting DNA from a subsample of 0.25 g using a Powersoil DNA

extraction kit (Qiagen) and sequenced as before. Further metagenomic assemblies from cryoconite hole sediment on Rotmoosferner glacier in the Ötztal Alps in Tirol, Austria[62] were mined for bacteriophage sequences using the VirSorter pipeline[63] primed with the Arctic 546 cryoconite virus contigs, retrieving a further 124 large virus contigs over 15 kb that were used in this study (Table 1).

**Metagenomic recruitment and MGI identification**. Virus contigs from meta-genomes Green, ML09 and AB09 (Table 1) were used to recruit reads from the other cryoconite virus-enriched metagenomes (<0.2 μm size fraction) plus a deeply sequenced Alpine cryoconite metagenome (ALP). FASTQ reads from each meta-genome were mapped to virus reference contigs using Bowtie2 v2.3[64] (settings: -very-sensitive, -X 0 -I 800 -no-unal). This aligned metagenomic reads >90% identity to the reference genomes. The resultant SAM files were imported into Geneious R8 (www. geneious.com) and manually viewed to check for large spikes, which may indicate error prone reads or unresolved repeat regions. These contigs were removed from further analysis to allow successful MGI detection with the methods below. The remaining mapped contigs which covered ≥70% of the reference were exported as BAM files for MGI detection. We did not recruit reads to genomes from their location of origin as the virus genomes we used were metagenomically assembled[34]. Meta-genomic assembly inherently selects for genomes where co-occurring MGI variants are low or undetectable, for example, if there are multiple competing MGI regions in a genome during assembly, an assembler will stop at the MGI region and output three or more contigs. We noted that many incomplete virus contigs in the original assembly appeared to be broken at MGI regions.

The genomecov module in BedTools v2.27[65] was used to calculate coverage for each base pair along each contig in the BAM file (bedtools genomecov -bga -ibam). MGIs were then identified according to the following criteria: In contigs with ≥5× coverage an MGI was predicted where coverage dropped to <25% of the mean over a ≥100 bp region; OR in contigs with 2–5× coverage, MGIs were identified where coverage dropped to zero over a ≥200 bp region. The coordinates of detected MGI regions were then extended a further 300 bp either side of the MGI to enhance identification of the gene containing the MGI using BLAST (Blast+ tools v2.6.0) and HMMER v3.1b searches, however only genes which directly overlapped with ≥10% of an MGI were considered for annotation. Extended coordinates were written as a BedFile and extracted as FASTA-formatted nucleotide sequences (BedTools getfasta)

**Variable gene, deletion event or intein**. To address whether MGIs in a gene are caused by variable gene sequences, whole gene deletion events or intein insertion/ deletions, we performed a variety of searches. For each MGI the corresponding SAM file was searched for discordantly mapping reads across the MGI region. Such reads were those which mapped greater than their expected paired-end read dis-tance and bridged the gap in read mappings, indicating a deletion event had occurred in this gene. Where discordantly mapping reads (>300 bp insert size) bridged the MGI with a coverage of >10% this was flagged as a deletion event in the query metagenome (either a whole gene deletion or that of an intein). To search for inteins present within MGI genes, all MGIs were searched against the NCBI conserved domain database (CDD) (https://www.ncbi.nlm.nih.gov/Structure/cdd/ wrpsb.cgi). CDD search results were filtered for known inteins and homing endonucleases. Where no discordant reads bridged the MGI, and no inteins were detected we assumed the MGI was caused by variable sequences within the gene.

**MGI annotation**. Genes were predicted on all extracted MGI regions using GeneMarkS v2.5[66] and the resulting nucleotide and amino acid sequences were annotated by searching against the GenBank non-redundant protein database (nr). (BLASTX, *e*-value cut-off 10⁻³), Pfam v29.0 (HMMScan *e*-value cut-off 10⁻⁵) and UniProtKB release 2016-6 (Phmmer *e*-value cut-off 10⁻⁷) (HH-suite v3).

**MGI gap-closing**. In two well covered genomes, a Mu-like bacteriophage and a novel Bacteroidetes phage, MGIs were closed in silico to determine the new sequence (if any) in each query metagenome (Supplementary Fig. 2). In summary, de novo-assembled contigs from the query metagenome were mapped to the reference contig and merged with short read mappings to form a new consensus genome. In detail and following Supplementary Fig. 2: (1) unassembled reads from each query metagenome were mapped to the virus reference sequence using Bowtie2 (settings: –very-sensitive -I 0 -X 800 -no-unal); (1b) The consensus sequence was generated in Geneious R8 (settings: 0% majority, Call Reference when no coverage). (2) Each individual query metagenome was trimmed using fastq mcf to remove Illumina adaptors, and de novo assembled using CLC assembly cell v4 with Kmer values of 23 or 33 and a bubble size of 2000 bp. (3) The query metagenomic assemblies were BLASTn searched (*e*-value cut off 10⁻⁵) against the consensus from step 2 and the resulting matching contigs were picked and imported to Geneious. (4) Matching contigs were again compared to the reference query using the Gen-eious mapper (settings: highest sensitivity, max number of gaps to 50%, max gap size 3 kb). Where the reference was a complete circular sequence, it was first cir-cularised in Geneious before mapping. The consensus genome (settings: 0% majority, reference excluded, call N when no coverage) was exported as a FASTA file. (5) To fill in any remaining gaps, unassembled reads from the query meta-genome were then mapped back to the new consensus sequence using the Geneious

read mapper (medium-low sensitivity) and allowed to iterate until complete. (6) Finally, to validate each newly reconstructed genomic variant, all mapped reads to the new variant were exported as paired reads, the reference genome was removed and the reads were assembled de novo in CLC assembly cell or Geneious R8 to produce the final circular genomes for each variant.

The entire process essentially allows targeted de novo assembly of individual bacteriophages from a metagenome by filtering out all other reads. All reconstructed virus genome variants were aligned with the original virus contig using Muscle v3.6. Genes were predicted on each virus variant in the alignment using Glimmer v3[67] and manually curated (Supplementary Fig. 1). For each MGI region, nucleotides from all variants were aligned using Muscle[68] and Maximum likelihood trees were constructed using PhyML v2.23[69] and scaled to size in Figs. 2 and 3.

**Detection of co-occurring MGI gene variants**. For each MGI in the reconstructed phages, reads from each metagenome were recruited to all variants, mapping only once to the closest match (bowtie2 -very-sensitive -I 0 -X 800 -no-unal). Where >70% of the reference MGI variant recruited reads at >1× coverage, a variant was flagged as being co-present in the metagenome. Its percentage coverage was calculated by dividing the mean coverage of each mapped variant present by the mean coverage of the core phage genome.

**Reporting summary**. Further information on research design is available in the Nature Research Reporting Summary linked to this article.

## Data availability

All raw and processed genomic data, including original and reconstructed virus genomes from each location (fasta and annotated GFF files) and all gene variants from each MGI are available on figshare: https://doi.org/10.6084/m9.figshare.11881773. The metagenomic island gapclosing protocol is available on protocols.io: dx.doi.org/10.17504/protocols.io.bix8kfrw. Raw metagenomic reads are available from the GenBank Sequence Read Archive (www.ncbi.nlm.nih.gov/sra) using the accession numbers in Table 1. Virus metagenomic assembled genomes used is this study are available from GenBank under the Bioproject PRJNA283341 (www.ncbi.nlm.nih.gov/bioproject). Cryophage and the Mu-like cryoconite phage have been deposited in GenBank under the accession numbers MT820023 and MT820024. Source data are provided with this paper.

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

## Acknowledgements

This work was supported by the Leverhulme Trust (RPG-2012-624) to A.M.A. and C.M.B. C.M.B. was also supported by the Austrian Science Fund (FWF, M 2299-B32). A.M.A. was also supported by NE/J02399X/1 and Aarhus University Research Funding (AuFF). A.E. was supported by NE/S001034/1.

## Author contributions

C.M.B., A.M.A., D.C.S. and G.B. devised the study, C.M.B. and A.E. performed sampling and metagenomic sequencing, C.M.B. devised the methodology and performed the bioinformatical analysis. C.M.B., A.M.A., G.B., A.E. and D.C.S. wrote the manuscript.

## Competing interests

The authors declare no competing interests.
