## [Peer Review File · Nature Communications]

Reviewers' comments:

Reviewer #1 (Remarks to the Author):

The manuscript describes the maintenance and variation in genomes of phages that inhabit cryonites. A very special environment generated by partial melting in pockets within glaciers. These environments are relatively simplified by their extreme conditions and effectively isolated, particularly when located at different and distant geographic sites. In this work, the viromes of cryonites in glaciers located at the Alps, Greenland and Svalvard have been analysed. They have found remarkable conservation in some of the genomes at the level of the core genome concurrent with a high diversity of flexible genome content at the level of localized genomic (metagenomic) islands. At the beginning of the manuscript the authors claim that they will be able to discriminate between the constant diversity model that predicts a high diversity at each site but little inter-site diversity and the arms-race model that would make different diversity to be found at the different locations due to divergent evolution. Although the results do support the first model the authors have developed little this important issue, particularly appropriate in the discussion that remains very superficial.

I believe the manuscript is an important contribution and should be published but some changes would be required.

Specifically:

Ln52. The term metagenomic islands was coined in reference 16 and was meant to describe regions of bacterial or archaeal genomes that underrecruited in metagenomes. The version found in phages was actually described in reference 10 and named metaviromic islands. It is all right to change names but it should be justified and properly explained. The reference PMID: 24616837 might help clarifying nomenclatures as well as the differences between expectations derived from CD and arms race.

Ln65. Refs 16 is the first describing the CD model

Ln101. Provide identity thresholds here (and in methods).

Ln114. What about genomes recruiting from the location of origin? Did they show MGIs?

Ln137. This part is very similar to the data described in ref 10. A description of what was found in marine phages in this paper and comparison with what has been found here seems appropriate.

Ln154. Please explain the term "additive MGI" and/or provide references

Ln171. The authors have focused the analysis into two specific genomes, there should be some kind of justification for this decision (completeness, coverage in different metagenomes etc.) and particularly refer to the comparison with other cases to assess how general the cases of those two phages can be considered.

Ln243 and figure 4. What are the threshold values for similarity to consider a read as covering a specific gene? (only GreenM is provided, 90% nucleotide identity to the reference)

Ln246. Horizontal gene transfer events are responsible regardless of the kind of recombination (homologous or site directed). In spite of previous evidence indicating homologous recombination to be the mechanism. Evidence for such a claim is not provided .

Ln263 I found the Discussion section rather poor in content. The evidence in favor of CD and against arms race as major drivers of phage (and cellular) evolution should be discussed as promised in the introduction and the advantages of the environment (cryonites) to effectively contribute to solve the conundrum highlighted here.

Reviewer #2 (Remarks to the Author):

Bellas et al report a metagenomic analysis of bacteriophage populations in cryconite holes, small water filled depressions in glaciers, from three locations in Svalbard, the Alps, and Greenland. The authors focus their work on metagenomic islands (MGIs) in abundant phage, and show that closely related phage often have MGIs that encode homologs with low nucleotide identity. This is attributed to a dynamic process of co-evolution with the host, in which homologous recombination plays a large role in maintaining diversity in phage populations.

I thought this paper was very interesting and contained some important insights into how genomic diversity is potentially maintained in bacteriophage populations. It is fascinating to observe that highly stable core phage genomes can be maintained over many years and thousands of kilometers, while hypervariable regions can be quickly exchanged, presumably to facilitate co-evolution with host defenses. The figures also do a nice job of conveying the major points. I am not an expert in cryo-microbiology, but this seems like an advance both in that field as well as to our broader understanding of bacteriophage biogeography and genomic diversity.

Major comments:

I was a bit confused regarding the methods for the generation of the sample-specific Mu and Bacteriodes phage genomes. It appears the authors have manually inspected the results carefully and the results appear robust, but given potential contig chimerism is an omnipresent problem with

these kinds of analyses I believe some clarification is warranted. Perhaps in Supplementary Figure 2 the authors could use different shapes for analyses vs data products, for example, and descriptions of the different consensus sequences generated (and how they differ) could be provided. It would also be useful if a general workflow was posted on GitHub or some other repository- I understand this isn't possible for the steps in Geneious, but for the other steps it would facilitate reproducibility.

The Mu-like phage in the ALP sample appears to have highly variable coverage, even in the conserved region (Figure 3). Are the authors sure this phage could be recovered completely in this sample? Perhaps re-mapping the reads with more stringent parameters might help, in case there are multiple closely related phage in that sample. It seems the --very-sensitive option of bowtie2 was used, but perhaps this maps reads from multiple populations.

Lines 59-72: I feel like the topics of phage/host microdiversity could be explained a bit more clearly. Since the focus of this study is not to disentangle the different mechanisms generating diversity it might be best to phrase this more as a statement of how mechanisms of generating diversity in phage have been postulated, yet paradoxically we still find very similar and stable phage genomes over space and time. I would also avoid the term "spread out", since it is not concise.

I would encourage the authors to discuss their findings in a bit more breadth to put their results in context. Right now the manuscript ends a bit abruptly and one feels there is a bit more to the story. For example, in the abstract the authors mention the interesting observation of how nearly identical phage genomes can recur in disparate times and locales, but background for this theme is not mentioned in the Introduction, nor is it revisited in the Discussion. In my opinion this is a critical point, since it is a major question in bacteriophage genomic diversity. I would encourage the authors to discuss their findings in light of some other studies that have made similar, seemingly paradoxical, observations, given the findings of this study provide a possible explanation and plausible mechanism for these patterns. There is a rich literature on these observations in marine systems that the authors could draw on to emphasize the potential significance of their findings, including the classic review piece "Here a virus.." (Breitbart and Rohwer, Trends in Micro., 2005) as well as some research articles:

Marston and Martiny, Environ Microbiol., doi: 10.1111/1462-2920.13556;

Pagarete et al., Appl. Env. Microbiol., DOI: 10.1128/AEM.01075-13,

Wommack et al., Appl. Env. Microbiol., 1999; [another classic]

Angly et al., PLOS Biology, 2006, doi.org/10.1371/journal.pbio.0040368

Roux et al., Nature, 2016 doi.org/10.1038/nature19366;

Aylward et al., PNAS, 2017, doi.org/10.1073/pnas.1714821114)

(I contributed to the last article, so I will leave it up to the authors to decide whether or not that warrants mention, but I believe the other articles at least should be mentioned in the context of how long-term stability of phage populations have been observed).

Minor comments:

Figure 1 caption: please provide abbreviation explanations- does AB refer to the Alps?

Line 15: perhaps a semicolon was intended here instead of a comma?

Line 16: not entirely clear what is meant by “analogous”. Perhaps re-phrase to something like “similar ecosystems that are geographically isolated”.

Line 46: change to “Indefinitely”

Line 57: some context would be useful here- consider rewording to “the abundant marine cyanobacterium *Prochlorococcus*...”

Line 276: perhaps the comma should be a period?

Figure 1 caption: capitalize Venn

Line 438: In -> in

Line 438-443: I’m assuming the authors checked to ensure entire genes were not encoded in the 300bp flanking regions? If so it is worth stating that here.

Line 470: I would avoid using the term “mapped” if the queries are not reads. If the queries are contigs it would be better to use the term “compared” or “searched”.

Please provide the version of Muscle, bowtie2, and other bioinformatic tools used.

Frank Aylward

Reviewers' comments:

Reviewer #1 (Remarks to the Author):

The manuscript describes the maintenance and variation in genomes of phages that inhabit cryonites. A very special environment generated by partial melting in pockets within glaciers. These environments are relatively simplified by their extreme conditions and effectively isolated, particularly when located at different and distant geographic sites. In this work, the viromes of cryonites in glaciers located at the Alps, Greenland and Svalbard have been analysed. They have found remarkable conservation in some of the genomes at the level of the core genome concurrent with a high diversity of flexible genome content at the level of localized genomic (metagenomic) islands. At the beginning of the manuscript the authors claim that they will be able to discriminate between the constant diversity model that predicts a high diversity at each site but little inter-site diversity and the arms-race model that would make different diversity to be found at the different locations due to divergent evolution. Although the results do support the first model the authors have developed little this important issue, particularly appropriate in the discussion that remains very superficial.

I believe the manuscript is an important contribution and should be published but some changes would be required.

Specifically:

Ln52. The term metagenomic islands was coined in reference 16 and was meant to describe regions of bacterial or archaeal genomes that underrecruited in metagenomes. The version found in phages was actually described in reference 10 and named metaviromic islands. It is all right to change names but it should be justified and properly explained. The reference PMID: 24616837 might help clarifying nomenclatures as well as the differences between expectations derived from CD and arms race.

Thank you for helping us clarify this. We have now correctly cited ref 16 for the first use of the term "metagenomic island". We decided to keep with the term "metagenomic island" to avoid confusion between metagenomes and metaviromes. Virus metagenomic islands are found by recruiting both metagenomes and metaviromes (<0.2 um) to reference genomes in our study. We now clarify this in L52-55.

PMID: 24616837 is indeed valuable to our discussion thank you. We have included this in our revised discussion on the CD model which we have significantly expanded.

Ln65. Refs 16 is the first describing the CD model

Changed accordingly thank you.

Ln101. Provide identity thresholds here (and in methods).

Bowtie2 works on a scoring threshold rather than an absolute identity threshold. A mismatched base at a high-quality position in the read receives a penalty of -6 by default. A length-2 read gap receives a penalty of -11 by default (-5 for the gap open, -3 for the first extension, -3 for the second extension). Thus, in end-to-end alignment mode, if the read is 100bp long and has 10 mismatches with no indels then this receives a score of $10 \times -6 = -60$

For an alignment to be considered "valid" (i.e. "good enough") by Bowtie 2, it must have an alignment score no less than the minimum score threshold. In end-to-end alignment

mode, this is calculated as $(-0.6 + (-0.6 * \text{Readlength}))$. So for a 100bp read this is -60. The minimum mapping identity with Bowtie2 is therefore 90% identity if there are no gaps. Therefore, we now write in the text > 90% identity.

Ln114. What about genomes recruiting from the location of origin? Did they show MGIs?

Yes and no, the original contigs used for recruitment were previously assembled from metagenomes. Metagenomic assembly of virus genomes from environmental samples positively selects for genomes where MGI variants are undetectable in the sample (or at least MGI regions are dominated by 1 variant). If there are multiple competing MGI regions in a genome during assembly, then the assembler will stop at the MGI region and output 3 or more separate contigs (unless the read lengths span the MGI, which is not the case for our Illumina data). In fact, many incomplete virus contigs appeared to be broken at MGI regions.

When a particular phage variant is highly dominant in a sample, the genome can be assembled in full without hindrance by the MGIs. This is why in Figure 4, Cryophage was assembled successfully from ML09, where one variant accounted for ~80% of read coverage, however it could not be *de novo* assembled in the other metagenomes, despite many of them having much higher coverages, because MGI variants were more evenly distributed in terms of abundance. We now add a paragraph into the Methods to state why we do not test for MGI within the sample of origin (L501-507 onwards).

Ln137. This part is very similar to the data described in ref 10. A description of what was found in marine phages in this paper and comparison with what has been found here seems appropriate.

We have now fully described the differences between the two datasets L193-199. Our dataset is largely in agreement, in that genes for host recognition are abundant in MGI regions. However, we found DNA methyltransferases were the most frequently transferred phage gene, presumably to counteract host RM systems. These could be both insertion/deletion events or variable methyltransferase genes. We also detected abundant homing endonucleases owing to a shorter MGI detection threshold.

Ln154. Please explain the term "additive MGI" and/or provide references

We mean a gene that is not variable, but is either present (added) or absent in different metagenomes. We now add this L180-181 "...meaning the gene or sequence could be present/absent but not variable."

Ln171. The authors have focused the analysis into two specific genomes, there should be some kind of justification for this decision (completeness, coverage in different metagenomes etc.)

Thank you for the comment, two specific genomes were chosen as these were two well covered complete genomes present across 5 or more metagenomes, with a variety of MGI regions. These genomes had the best chance of allowing MGI reconstructions (which relies on having sufficient coverage of each variant in each metagenome).

We now state this in L208-211. MGIs. We use these two unrelated genomes to show the fine scale details and relationships between the flexible phage genes. The detection of MGIs on a larger scale, across multiple phage types, is used to show these events are common across all phage genomes.

Ln243 and figure 4. What are the threshold values for similarity to consider a read as covering a specific gene? (only GreenM is provided, 90% nucleotide identity to the reference)

We now state in both locations that the threshold is >90 % nucleotide identity.

Ln246. Horizontal gene transfer events are responsible regardless of the kind of recombination (homologous or site directed). In spite of previous evidence indicating homologous recombination to be the mechanism. Evidence for such a claim is not provided .

We have changed this to just state "recombination events" are responsible

Ln263 I found the Discussion section rather poor in content. The evidence in favor of CD and against arms race as major drivers of phage (and cellular) evolution should be discussed as promised in the introduction and the advantages of the environment (cryonites) to effectively contribute to solve the conundrum highlighted here.

Thank you for your suggestions on improving the discussion. We agree the evidence does support the model that a constant diversity of phages exist in each location and that arms-race dynamics appear to have a more limited contribution to generating diversity over the spatial and temporal scales used in the study. We address this in a fully revised discussion (particularly L324-339), but also add that whilst CD must be driving the large scale maintenance of diversity, arm-race dynamics may still be occurring on the scale of single nucleotide variations, as they are not mutually exclusive processes. However, the detection of similar phage receptor genes thousands of km apart suggests arms race dynamics do not create unlimited diversity, It may be subject to fluctuating selection.

We have also highlighted the advantage of using cryoconite in this, and future, studies of microbial diversity (L304-308). Thank you for this suggestion.

Reviewer #2 (Remarks to the Author):

Bellas et al report a metagenomic analysis of bacteriophage populations in cryoconite holes, small water filled depressions in glaciers, from three locations in Svalbard, the Alps, and Greenland. The authors focus their work on metagenomic islands (MGIs) in abundant phage, and show that closely related phage often have MGIs that encode homologs with low nucleotide identity. This is attributed to a dynamic process of co-evolution with the host, in which homologous recombination plays a large role in maintaining diversity in phage populations.

I thought this paper was very interesting and contained some important insights into how genomic diversity is potentially maintained in bacteriophage populations. It is fascinating to observe that highly stable core phage genomes can be maintained over many years and thousands of kilometers, while hypervariable regions can be quickly exchanged, presumably to facilitate co-evolution with host defenses. The figures also do a nice job of conveying the major points. I am not an expert in cryo-microbiology, but this seems like an advance both in that field as well as to our broader understanding of bacteriophage biogeography and genomic

diversity.

Major comments:

I was a bit confused regarding the methods for the generation of the sample-specific Mu and Bacterioidetes phage genomes. It appears the authors have manually inspected the results carefully and the results appear robust, but given potential contig chimerism is an omnipresent problem with these kinds of analyses I believe some clarification is warranted. Perhaps in Supplementary Figure 2 the authors could use different shapes for analyses vs data products, for example, and descriptions of the different consensus sequences generated (and how they differ) could be provided. It would also be useful if a general workflow was posted on GitHub or some other repository- I understand this isn't possible for the steps in Geneious, but for the other steps it would facilitate reproducibility.

Thank you for carefully following the workflow and for suggesting clarifications. It is our aim to make this robust and reproducible. To this end we have deposited an in depth work-flow on protocols.io (This will be made public upon publication).

<https://www.protocols.io/private/6EF3C82E8B0011EA9D080A58A9FEAC2A>

We have also added a key to Supplementary Figure 2 and clarified in the legend that analysis steps are in white, data products are coloured rectangles (light blue referring to metagenomic reads, yellow is the virus reference genome).

The entire analysis workflow essentially filters metagenomic reads from the query metagenome, so that only those mapping to the virus reference contig remain (including the missing regions in the MGIs). We remove the reference and then *de novo* assemble these reads as an additional check to confirm the new genomes.

We have also updated the data availability statement to include a link to all raw and processed data, including original and reconstructed virus genomes from each locations (fasta and gff).

The Mu-like phage in the ALP sample appears to have highly variable coverage, even in the conserved region (Figure 3). Are the authors sure this phage could be recovered completely in this sample? Perhaps re-mapping the reads with more stringent parameters might help, in case there are multiple closely related phage in that sample. It seems the --very-sensitive option of bowtie2 was used, but perhaps this maps reads from multiple populations.

Thank you for the comment and suggestions, the variable coverage arises because the ALP genome contains a few more MGIs than the others. The read mappings in Figure 3 are before the genome reconstructions, so these MGIs show up as additional gaps and unevenness in the sample (including two small variable genes in the 'core region' - we only analysed MGIs in detail when they were across two or more samples). We are confident about the genome reconstructions as the ALP sample in this particular phage appeared to be dominated by 1 variant (>80% of reads mapped to one variant in the RBP, Terminase and Tail tape measure gene - Figure 3) which actually allowed for *de novo* assembly from the ALP metagenome of large contigs, these could be aligned to the original (AB09) virus scaffold and plug the gaps.

Lines 59-72: I feel like the topics of phage/host microdiversity could be explained a bit more clearly. Since the focus of this study is not to disentangle the different mechanisms generating diversity it might be best to phrase this more as a statement of how mechanisms of generating diversity in phage have been postulated, yet paradoxically we still find very similar and stable phage genomes over space and time. I would also avoid the term “spread out”, since it is not concise.

Thank you for these comments. We have clarified this section and now stated that these mechanisms have been postulated to generate phage diversity. We remove the term spread out. This section now feeds into a new paragraph on the paradox of the known rapidly changing nature of phage genomes, yet stable genomes are observed across the oceans (L80-87).

I would encourage the authors to discuss their findings in a bit more breadth to put their results in context. Right now the manuscript ends a bit abruptly and one feels there is a bit more to the story. For example, in the abstract the authors mention the interesting observation of how nearly identical phage genomes can recur in disparate times and locales, but background for this theme is not mentioned in the Introduction, nor is it revisited in the Discussion. In my opinion this is a critical point, since it is a major question in bacteriophage genomic diversity. I would encourage the authors to discuss their findings in light of some other studies that have made similar, seemingly paradoxical, observations, given the findings of this study provide a possible explanation and plausible mechanism for these patterns. There is a rich literature on these observations in marine systems that the authors could draw on to emphasize the potential significance of their findings, including the classic review piece

“Here a virus..” (Breitbart and Rohwer, Trends in Micro., 2005) as well as some research articles:

Marston and Martiny, Environ Microbiol., doi: 10.1111/1462-2920.13556;

Pagarete et al., Appl. Env. Microbiol., DOI: 10.1128/AEM.01075-13,

Wommack et al., Appl. Env. Microbiol., 1999; [another classic]

Angly et al., PLOS Biology, 2006, doi.org/10.1371/journal.pbio.0040368

Roux et al., Nature, 2016 doi.org/10.1038/nature19366;

Aylward et al., PNAS, 2017, doi.org/10.1073/pnas.1714821114)

(I contributed to the last article, so I will leave it up to the authors to decide whether or not that warrants mention, but I believe the other articles at least should be mentioned in the context of how long-term stability of phage populations have been observed).

Thank you for these suggestions on improving the discussion and providing some references to do this, we also agree that understanding how stable phage genomes are globally maintained is a major question in phage ecology. We now expand this area by providing an additional paragraph in the introduction using the above references to give examples of long-term and global re-occurrence of virus genomes (L80-87 onwards) and the paradox this creates with the known genetic exchange and mutation in phages. We revisit this in a fully revised discussion (L341-363) to show that the MGIs we reconstruct in cryoconite holes provides a plausible explanation for these observations. Briefly, we see stable, widespread, core phage genomes are maintained in every location, but with variable MGI regions at specific locations on the genomes which create diversity where it counts (host recognition and countering bacterial defences). Hence, over 90% of a phage

genome is 99% identical, meaning it would be detected as the same phage in many of the earlier studies described. However, the variable genes create the necessary microdiversity to infect diverse host populations.

Minor comments:

Figure 1 caption: please provide abbreviation explanations- does AB refer to the Alps?

Abbreviations added thank you

Line 15: perhaps a semicolon was intended here instead of a comma?

Changed

Line 16: not entirely clear what is meant by “analogous”. Perhaps re-phrase to something like “similar ecosystems that are geographically isolated”.

Changed, thank you

Line 46: change to “Indefinitely”

Corrected

Line 57: some context would be useful here- consider rewording to “the abundant marine cyanobacterium Prochlorococcus...”

Changed as advised

Line 276: perhaps the comma should be a period?

Discussion fully revised

Figure 1 caption: capitalize Venn

Corrected

Line 438: In -> in

Corrected

Line 438-443: I’m assuming the authors checked to ensure entire genes were not encoded in the 300bp flanking regions? If so it is worth stating that here.

We did check for flanking genes, but this was stated two paragraphs later in the MGI annotation section, however we now move this forward to improve the clarity (L517-518) "Only genes that directly overlapped with >10% of an MGI region were considered for annotation"

Line 470: I would avoid using the term “mapped” if the queries are not reads. If the queries are contigs it would be better to use the term “compared” or “searched”.

Corrected

Please provide the version of Muscle, bowtie2, and other bioinformatic tools used.

Added to Methods now

Frank Aylward

REVIEWERS' COMMENTS:

Reviewer #1 (Remarks to the Author):

The manuscript can be accepted as it is

Reviewer #2 (Remarks to the Author):

The authors have addressed my concerns.

One small comment from reading the revised section in the Discussion-

Line 355-358- please add appropriate citation here (not clear from the context).

Reviewer #3 (Remarks to the Author):

I was specifically asked to review the cryoconite holes section and i note that author have missed some literature on viruses in cryoconite hole as well some of the bacterial dynamics papers. I bring this up as the authors expand the groundwork for phage-bacterial dynamics from their previous work. The research community in the Antarctic is also studying microbial (inc. viruses) cryoconite holes (PMID: 30778338; 31689942, 29228256).

As side note for future: This study has heavily relied on read recruitment to known phages. This means that any unknown viral signatures will be missed.

I see no major issue with the cryoconite holes section.

REVIEWERS' COMMENTS:

Reviewer #1 (Remarks to the Author):

The manuscript can be accepted as it is

Reviewer #2 (Remarks to the Author):

The authors have addressed my concerns.

*One small comment from reading the revised section in the Discussion-
Line 355-358- please add appropriate citation here (not clear from the context).*

missing citation added thank you (now line 354)

Reviewer #3 (Remarks to the Author):

I was specifically asked to review the cryoconite holes section and i note that author have missed some literature on viruses in cryoconite hole as well some of the bacterial dynamics papers. I bring this up as the authors expand the groundwork for phage-bacterial dynamics from their previous work. The research community in the Antarctic is also studying microbial (inc. viruses) cryoconite holes (PMID: 30778338; 31689942, 29228256).

Thank you for directing us towards relevant antarctic studies. We have now included referencing to the above in the section on cryoconite holes

As side note for future: This study has heavily relied on read recruitment to known phages. This means that any unknown viral signatures will be missed.

The phage "reference" genomes were originally assembled from the same metagenomes we use for the read mappings (see PMID: 26191051), so we are including novel virus diversity in this analysis. However, we agree future studies must use metagenomes where the phage genome is thought to be present, we now clarify this in L302-304.

I see no major issue with the cryoconite holes section.